# Geometric Versus Anemometric Surface Roughness for a Shallow Accumulating Snowpack

**Jessica E. Sanow [1,2], Steven R. Fassnacht [2,3,4,\*], David J. Kamin [3,5], Graham A. Sexstone [6], William L. Bauerle [7] and Iuliana Oprea [8]**

1   Geosciences, Colorado State University, Fort Collins, CO 80523-1482, USA; Jessica.Sanow@colostate.edu
2   Natural Resources Ecology Laboratory, Colorado State University, Fort Collins, CO 80523-1499, USA
3   ESS-Watershed Science, Colorado State University, Fort Collins, CO 80523-1476, USA; kamind@gmail.com
4   Cooperative Institute for Research in the Atmosphere, Colorado State University, Fort Collins,
     CO 80523-1375, USA
5   Now with Xcel Energy, 1800 Larimer St, Denver, CO 80202, USA
6   Colorado Water Science Center, U.S. Geological Survey, Lakewood, CO 80225, USA; sexstone@usgs.gov
7   Horticulture and Landscape Architecture, Colorado State University, Fort Collins, CO 80523, USA;
     Bill.Bauerle@colostate.edu
8   Mathematics, Colorado State University, Fort Collins, CO 80523, USA; Juliana@math.colostate.edu
\*   Correspondence: Steven.Fassnacht@colostate.edu

**Abstract:** When applied to a snow-covered surface, aerodynamic roughness length, $z_0$, is typically considered as a static parameter within energy balance equations. However, field observations show that $z_0$ changes spatially and temporally, and thus $z_0$ incorporated as a dynamic parameter may greatly improve models. To evaluate methods for characterizing snow surface roughness, we compared concurrent estimates of $z_0$ based on (1) terrestrial light detection and ranging derived surface geometry of the snowpack surface (geometric, $z_{0G}$) and (2) vertical wind profile measurements (anemometric, $z_{0A}$). The value of $z_{0G}$ was computed from Lettau's equation and underestimated $z_{0A}$ but compared well when scaled by a factor of 2.34. The Counihan method for computing $z_{0G}$ was found to be unsuitable for estimating $z_0$ on a snow surface. During snowpack accumulation in early winter, $z_0$ varied as a function of the snow-covered area (SCA). Our results show that as the SCA increases, $z_0$ decreases, indicating there is a topographic influence on this relation.

**Keywords:** aerodynamic roughness length; terrestrial lidar; snow surface topography; wind profile; snow energy balance; snow accumulation

## 1. Introduction

In the Northern Hemisphere, a seasonal snowpack can cover over 50% of the land area with the snow surface often the interface between the atmosphere and the earth [1]. The roughness of a snow surface is an important control on air-snow heat transfer [2], and changes in the snow surface can have substantial effects on the energy balance at this interface. Snow is a complicated surface with rapidly evolving physical roughness characteristics due to changing atmospheric conditions, the metamorphism of snow crystals, melting and freezing processes and redistribution by wind, especially in open areas [3]. Roughness characteristics also influence the air-surface momentum transfer on the snowpack due to wind [4]. The changes in wind momentum can reduce the energy budget, influence the formation of roughness features, and affect the aeolian movement of snow [4]. Heat flux modeling has typically used the aerodynamic roughness length ($z_0$) as a static parameter, in hydrologic, snowpack, and climate models [5,6], with $z_0$ only varying as a function of land cover type. For example, the Community Land Model version 4.0 (CLM4; http://www.cesm.ucar.edu/models/ccsm4.0/clm/)

applies a single $z_0$ value of $2.4 \times 10^{-3}$ m to all snow-covered surfaces. However, $z_0$ varies both spatially [7] and temporally [8], which may result in variable estimates of turbulent heat fluxes not captured by most models [9]. Wind velocity profile measurements are often used to calculate $z_0$ estimates [6], but there are a limited number of sites that measure the wind profile over a snowpack surface, making the spatio-temporal representation of $z_0$ challenging.

Millimeter-scale variations in snow-surface roughness features have been estimated from a black plate pushed partially into the snow [10–12], using two-dimensional photography, digital processing, and automated post-processing software [13–16]. Snow surface elevation data are now available over large areas at the resolution ($\pm 80$ mm) of airborne light detection and ranging (lidar) [17–19], terrestrial laser scanner (TLS) (resolution of $\pm 5$ mm) [20–26], and photogrammetry [25]. Although most lidar and photogrammetry efforts have only focused on snow depth [26], only a few datasets have been used to evaluate snow surface roughness at the meter-scale or sub-meter scale [27,28]. However, few of these datasets have been applied to interpolate $z_0$ and create a digital elevation model of the snowpack surface for evaluating surface roughness [27]. Aerodynamic roughness length ($z_0$) has been estimated from the geometry of the snow surface [2,7,29–31]. However, this method is time consuming and typically only applicable over smaller scales [13]. Also, Fassnacht et al. [27] have identified potential errors with the different methods of computing $z_0$ from the geometry of the surface that result in values varying over 1–3 orders of magnitude and have suggested these methods need to be evaluated for varying scales, resolutions, and environments.

This study used TLS-derived surface geometry and vertical wind profile measurements to compare concurrent $z_0$ estimates for changing snow surface features of shallow snowpacks. Here, we asked the following questions: (1) How does the aerodynamic roughness length ($z_0$) vary spatially and temporally for a shallow snow environment? (2) How does $z_0$ estimated from geometric measurements ($z_{0G}$) compare to $z_0$ estimated from anemometric measurements ($z_{0A}$), and (3) How does $z_0$ vary with snow-covered area based on the underlying terrain?

## 2. Materials and Methods

The capability of a rough surface to absorb momentum from a turbulent boundary layer can be quantified by $z_0$, which is a measure of the vertical turbulence that occurs when a horizontal wind flows over a rough surface [32]. In general, $z_0$ is a quantity that is computed from the Reynolds number and the roughness geometry of the surface [29]. For rough, turbulent regimes occurring in the atmospheric boundary layer, dependence on the Reynolds number vanishes and $z_0$ is only a function of roughness geometry [33]. Various relations have been found to relate the geometry of roughness elements with $z_0$ [2,29]. For example, the dependence of $z_0$ on the size, shape, density, and distribution of surface elements has been studied using wind tunnels, analytical investigations, numerical modeling, and field observations [34,35]. Smith [36] provides a comprehensive review of the different approaches and models developed to analyze surface roughness and highlights that almost all models were developed for simplistic natural surfaces (i.e., regular arrays of roughness elements).The lack of a clear method for calculating $z_0$ as a function of surface roughness is due to the complexity of surfaces that exists in nature and the direction, spatial, and temporal dependence.

The most robust approach for estimating $z_0$ is from the anemometric method used to generate a logarithmic wind profile and solve for $z_0$ [32]. The anemometric method can be used for any surface with any arrangement of roughness elements but requires a meteorological tower of at least two vertically spaced wind, temperature, and humidity measurements that can be used to approximate the respective gradients. The measurements integrate over a footprint area rather than the single-point location of the sensors based on the distance from measurement source, elevation of sensor, meteorological conditions, turbulent boundary layer, and atmospheric stability. All of these factors can potentially create turbulent fluctuations affecting the downwind measurements of the wind profile [37,38]. The anemometric method is also very sensitive to the wind measurement heights; Munro [2] found that adding 0.1 m to any of the heights can alter $z_0$ by an order of magnitude.

In contrast, the geometric method uses algorithms relating $z_0$ to characteristics of surface roughness elements and thus does not require tower instrumentation but only a measure of the geometry of the surface [29].

Anemometric data are used to estimate $z_0$ from the logarithmic wind profile through an empirical relation that describes the vertical distribution of horizontal wind speeds within the lowest portion of the planetary boundary layer [39]. The wind speed ($U_z$ in m/s) at height $z$ (in m) above a surface is given by:

$$U_z = \frac{U^*}{k} \ln\left[\frac{z}{z_0} + \psi\left(z, \frac{z}{L}, L\right)\right] \tag{1}$$

where $U^*$ is the friction velocity (m/s), k is the Von Kármán constant (~0.40), and $\psi$ is a stability term, and $L$ is the Monin-Obukhov stability parameter. This equation is only valid through the hypothesis of stationarity and horizontal homogeneity. Under neutral stability conditions, $z/L$ tends towards zero, and $\psi$ can be neglected.

The most common geometric method for estimating $z_0$ is simply a function of the height of the elements:

$$z_0 = f_0 z_h \tag{2}$$

where $z_h$ is the mean height of roughness elements in meters, and $f_0$ is an empirical coefficient derived from observation [28]. The frontal area index, which combines mean height and breadth (all in meters), and density of the roughness elements, is defined as roughness area density given by:

$$(\lambda F) = Ly\, z_h\, \rho el \tag{3}$$

where $Ly$ is the mean breadth of the roughness elements perpendicular to the wind direction, and $\rho el$ is the density or number ($n$) of roughness elements per unit area [40]. Lettau [29] developed a formula for $z_0$ based on the geometry of the surface for irregular arrays of reasonably homogenous elements:

$$z_0 = 0.5\, z_h\, \lambda\, F \tag{4}$$

In the Lettau formula, the coefficient 0.5 represents an average drag coefficient for the roughness elements, which was determined experimentally. Other geometric methods have been developed, especially to consider more regularly-shaped and distributed roughness elements, such as buildings in an urban setting [41,42]. The Counihan equation provides a geometric estimate of $z_0$ as:

$$z_0 = z_h\left(1.8\, \frac{A_f}{A_d} - 0.08\right) \tag{5}$$

where $A_f$ is the total area in square meters silhouetted by the roughness elements, and $A_d$ is the total area covered by roughness elements.

A meteorological tower was erected at Colorado State University Agricultural Research, Development and Education Center (ARDEC) South (http://aes-ardec.agsci.colostate.edu/), (40.629680, −104.99699) on a flat field that had no obstructions at least 100 m in the prevailing wind direction. The fetch was 40 m wide with the tower placed in the middle, leaving 100 unobstructed, homogenous meters upwind. Ten anemometers and five temperature and relative humidity sensors were placed vertically at different heights on the tower. The accuracy of the air temperature and relative humidity sensors (METER VP-3) was variable across a range of ±0.25–0.50 °C and ±4%, respectively (see http://manuals.decagon.com/Manuals/14053_VP-3_Web.pdf for more information). The METER Davis Cup Anemometers have a wind direction accuracy of ±7° and a speed accuracy within ±5% (see http://manuals.decagon.com/Manuals/). Data were collected from February 2014 through March 2015. In mid-March 2014, the flat field was plowed to create additional underlying roughness, specifically furrows and troughs were formed perpendicular to the dominant wind direction at an

approximate spacing of two meters. The approximate amplitude of the troughs and furrows was 25 cm deep and 50 cm wide.

Meteorological data were recorded every five minutes based on the average of one-minute observations. Anemometric data were evaluated for 153 instances when wind speeds were faster than 4 m/s to ensure neutral stability [8] and when the log-linear fit had an $r^2$ greater than 0.95. The height of the instruments was calculated based on the depth of snow, which did not exceed 10 cm.

This study estimated $z_0$ values from anemometric measurements and used them as a reference to evaluate concurrent geometric methods. The $z_{0A}$ values were calculated using Equation (1) from logarithmic anemometer wind profile data. Surface elevation was measured using a FARO Focus3D X 130 model TLS (https://www.faro.com/products/). This lidar tool generates a point cloud scan of a given area with an error of $\pm 2$ mm and a resolution of approximately 7.5 mm. The TLS was set up in 2–3 locations around the area of interest with 6 reference spheres to match the images using the FARO Scene Software. The data were generated into a point cloud and interpolated to a solid surface with 10 mm resolution with the kriging method using the Golden Software's Surfer 8 (https://www.goldensoftware.com/products/surfer). The gridded data were de-trended in the x-y plane to remove the bias in slope of the field or the angle of the lidar. Gaps in the scans tended to be small (<100 mm), and the kriging interpolation eliminated them. Individual roughness elements were identified and for each element the silhouette lot area and obstacle height were determined using a MATLAB code (https://www.mathworks.com/products/matlab.html). This was required to compute the Lettau formula (Equation (4)). The 1000-$m^2$ area around the tower was scanned on 12 occasions when the concurrent anemometric and geometric measurements were acquired. One concurrent measurement set was made with no snow cover for each of the unplowed and plowed scenarios; seven concurrent measurement sets were made with partial snow-covered area (SCA) and three with complete snow cover. SCA was determined from digital photos taken from the TLS unit.

Both the Counihan and Lettau methods were used to calculate $z_{0G}$ (Equations (4) and (5), respectively). The Counihan method was appropriate for this study because the roughness elements (furrows) in this study site were semi-regular. During each concurrent anemometric and geometric measurement set, the percentage of the area covered in snow, or SCA, was estimated from photographs.

## 3. Results

The unplowed versus plowed field yielded different $z_{0A}$ values (Figure 1). On average, the plowed field was almost 20 times as rough as the unplowed field, yet the coefficient of variation (COV) was essentially the same (0.67 and 0.62, respectively) (Figure 1). The smallest $z_{0A}$ values for the plowed field were of the same magnitude as some of the largest $z_{0A}$ values for the unplowed field, in the range of 1 to $3 \times 10^{-3}$ m.

The Counihan method estimated $z_{0G}$ values that were 1.39 times larger and had greater variation than the estimated $z_{0A}$ values (Figure 2). We used the Nash-Sutcliffe coefficient of efficiency (NSCE), which is a performance statistic based on a comparison of the data fit to the 1:1 line, to evaluate how estimates of $z_{0G}$ compared with $z_{0A}$ [43]. The NSCE of the Counihan $z_{0G}$ was $-1.18$, and the Lettau $z_{0G}$ was 0.14, indicating the Lettau method compared more favorably with the $z_{0A}$. A linear regression between both $z_{0G}$ estimates (Counihan $z_{0G}$ and Lettau $z_{0G}$) and $z_{0A}$ was fit through the data origin to evaluate if the bias between the two methods could be removed through simple linear scaling (Figure 2). When the Counihan $z_{0G}$ values were scaled by 0.721 (1/1.39), the NSCE value only increased to 0.07. However, the NSCE increased to 0.88 when the Lettau $z_{0G}$ values were scaled by 2.34 (1/0.428).

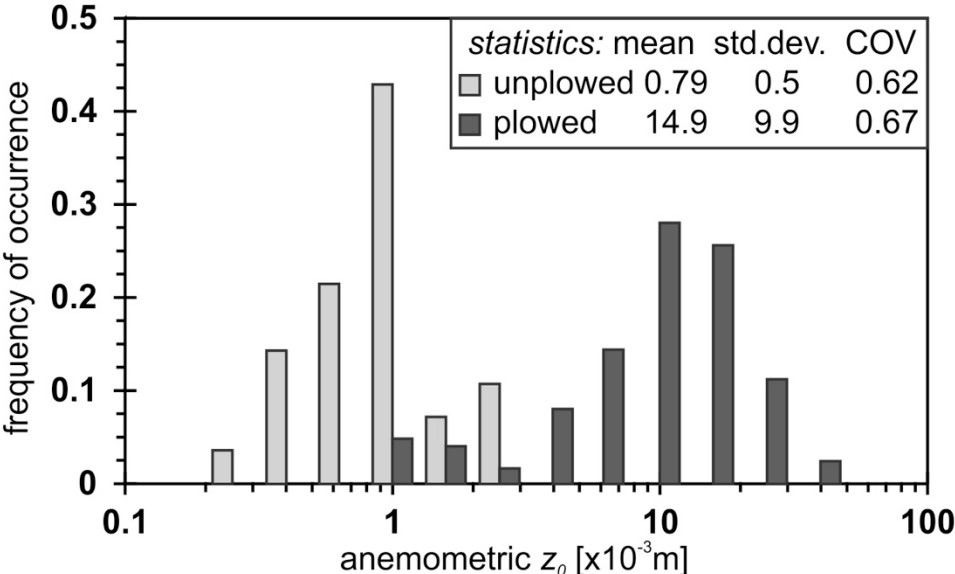

**Figure 1.** Histogram showing range and distribution of anemometric $z_0$ ($z_{0A}$) values for the unplowed and plowed field from anemometric data, based on 28 and 125 wind-speed profiles, respectively. The summary statistics (mean, standard deviation (std.dev.), and coefficient of variation (COV)) are presented in the legend. A logarithmic scale is shown on the x-axis to highlight the large difference for $z_{0A}$ values among fields with varying characteristics.

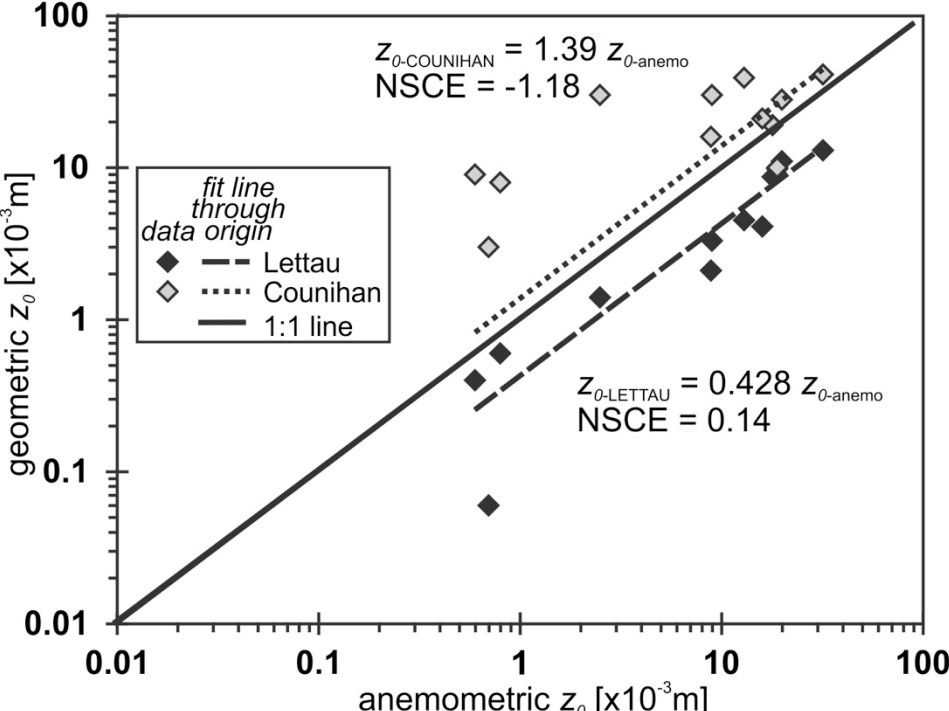

**Figure 2.** Comparison of Lettau and Counihan geometric methods to the anemometric method. A linear regression between the geometric-based Lettau and Counihan methods ($z_{0G}$) and the anemometric method ($z_{0A}$) fit through the origin are presented. The Nash-Sutcliffe coefficient of efficiency [43] fit statistic is also presented. When the Lettau method is scaled by 2.34 (1/0.428), the NSCE increases to 0.88. For the Counihan comparison, when it is scaled by 0.721 (1/1.39), NSCE increases to 0.07.

The estimated $z_0$ values were found to vary as a function of the amount of SCA present (Figure 3). As SCA increases, $z_0$ decreases, with variability based on the calculation method (Figure 3a). A linear

regression between SCA and each of the $z_0$ estimates showed $r^2$ values that were 0.01, 0.7, and 0.88 for the Counihan, anemometric, and Lettau methods of $z_0$ calculation, respectively. There were noticeable differences in $z_0$ depending whether SCA was increasing because snow was accumulating versus when SCA was decreasing because the snow was melting. For periods of snow accumulation, removing snow measurements that were not immediately following a snow event (the yellow boxes in Figure 3b that represent non-accumulation values) improved the linear relation between accumulating SCA and $z_0$ ($R^2 = 0.94$).

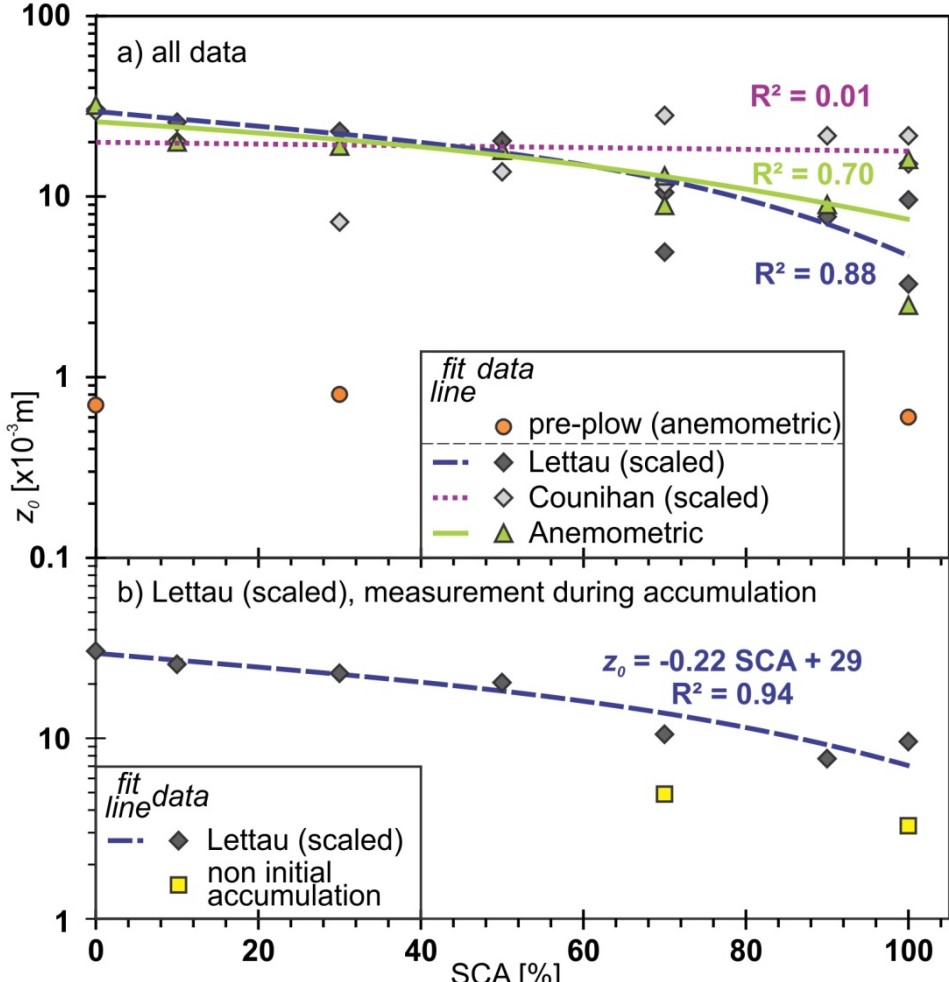

**Figure 3.** Linear relation between $z_0$ and snow-covered area (SCA as a %) for (**a**) all datasets (scaled Lettau and Counihan geometry-based and anemometric-based) with anemometric-based $z_0$ for the pre-plowed (orange circles) and plowed fields (green triangles) highlighted, and (**b**) the scaled Lettau geometry-based $z_0$ using a factor of 2.34 (see Figure 2). Lettau geometry-based $z_0$ measurements with non-accumulation snow measurements were removed. Lines are based on the best-fit linear regression of the data. Snow had been on the ground for numerous days prior to the two concurrent measurements (yellow squares) taken on 22 March 2014 (SCA = 100%) and 13 April 2014 (SCA = 70%). The snowfall was fresh for all other measurements.

## 4. Discussion

Geometrically estimated $z_0$, although easier to measure, produced different values when compared to the anemometric derived values. The Counihan method overestimated by a factor of 0.721, whereas the Lettau method underestimated anemometric $z_0$ (Figure 2) by a factor of 2.34. The Lettau method (Equation (4)) has a constant of 0.5 based on the average drag coefficient of the roughness characteristic of the silhouetted area of the average obstacle. By dividing the Lettau based

$z_0$ values by the 0.5 and thus eliminating the drag coefficient from the equation, we get a new NSCE of 0.856, with scatter in the data much closer to the 1:1 line (Figure 2). The removal of the drag coefficient suggests that the geometric data generated from the lidar point cloud appears to account for spatial and temporal variability in the roughness of a snow surface.

Lidar-based snow data are becoming more readily available [19,26]. The accuracy of the scans from about 1 mm for terrestrial lidar to 10 cm for airborne lidar can account for fine-scale changes of the snowpack [26], which enables the computation of $z_0$ at any scale. Although anemometric data can yield reliable estimates of $z_0$, meteorological towers are expensive to set up and operate. In addition, data from a single tower does not consider spatial variability as well as the geometric method [44]. Comparing the two methods does not consider the scale of the study area; the geometric measurement is taken over the entire area near the meteorological tower whereas the anemometric measurement is only influenced by the fetch area upwind of the sensors [29].

The roughness of the snowpack can vary substantially both spatially and temporally creating many implications [13,14,45]. Roughness variations can be caused by heterogeneities in land cover, vegetation, and meteorological conditions [46]; non-uniform distribution of snow cover during accumulation and melt [45,46]; snow-canopy interactions [47]; and snow redistribution by wind [48]. This was apparent in differences between the estimated $z_0$ for the plowed versus unplowed field (Figure 1). Land cover varies throughout regions particularly those with a shallow snow environment, and this creates variations that depend on the underlying topography [13,14,46]. Thus, there are many different values of $z_0$ in the literature [7] that are broader than our observed mean range of 0.2 to $10 \times 10^{-3}$ m (Figure 1). For example, Miles et al. [31] found the $z_0$ of a hummocky glacier (a particularly rough underlying surface) to range between 5 to $500 \times 10^{-3}$ m, whereas Brock et al. [7] reported $z_0$ values for fresh snow and older snow of $0.2 \times 10^{-3}$ and $3.56 \times 10^{-3}$ m, respectively. Our results show that change in roughness between a plowed and unplowed field yielded a 20-fold difference in $z_0$. The results of this study can be applied to areas of similar climate and land cover, which included flat, bare soil, and bare soil with small furrows (<1 m); and therefore, the results of this study may not scale appropriately to different land cover types. Further studies of a shallow snowpack in sagebrush steppe [49], farmland, or non-densely forested environments may be able to replicate our study results and scale from 1000 $m^2$ to a larger area. The $z_0$ values observed here had a notable change between flat soil and small furrows, so the changes in $z_0$ values in different environments with even minimal vegetation will have much larger effects on the $z_0$ values.

The inverse relation of SCA and $z_0$ (Figure 3) [50] is affected by the underlying terrain and size of the roughness features. As the snow accumulation increases, the roughness elements become buried, and the topography appears to be smooth [50,51]. This relation indicates that as snow accumulates over topographic features the snow will begin to level out at a $z_0$ height dependent scale. A hysteresis can be noted, and it has been found that a single snowfall event on a hummocky glacier can alter the micro-topography by up to 75% due to the shallow snowpack over the small scaled features [45,52]. The CLM4 uses a $z_0$ value of $2.4 \times 10^{-3}$ m, a value that falls near the mean of the unplowed field, which is applicable for deep, flat snowpack surface with minimal influences from underlying terrain. However, this is not typical for shallower snowpacks or in complex terrains.

Relations between $z_0$ and SCA (Figure 3) can be used to improve snow-energy balance modeling by estimating the percentage of SCA via remote sensing and applying $z_0$ only to the portion of area it accurately describes [46,53]. Currently, most models use 100% SCA even though many areas will remain snow free due to complex terrain and can drastically change during periods of melt and accumulation [13,53]. Aerodynamic roughness length is incorporated into many climate and energy models, which require sub-grid snow distribution [54] and are still inadequate at representing SCA [46,48]. A dynamic $z_0$ based on SCA and land cover type can improve these on a sub-grid scale. Another complication with these models is the lack of accountability for snowpack variability throughout accumulation and melt [48,53,54].

Resolution is an important factor to consider when discussing both SCA and $z_{0G}$. The higher the resolution of the measurements (lidar, satellite, etc.), the higher the $z_{0-G}$ accuracy. However, lidar datasets are often large, especially those acquired with TLS, making them difficult and time consuming to process. Lower resolution data from remote sensing or airborne lidar systems (ALS) can cause problems when scaling [53]. Quincey et al. [52] found that $z_{0G}$ is typically underestimated with a small area and coarse resolution and overestimated with a large area and fine resolution when compared to anemometric data. Nonetheless, even with lower resolution, applying dynamic $z_0$ values may greatly improve models. Scaling can be an effective way to incorporate both an anemometric and geometric $z_0$ value. Based on a specific land cover type, a scaling factor can be applied to areas with the same land cover. This can help to improve modeled $z_0$ accuracy, once preliminary $z_0$ values have been established.

## 5. Conclusions

Aerodynamic roughness length within our study has shown variation spatially and temporally for a shallow accumulating snow environment. This was apparent in our results that showed differences in $z_0$ mean values of about $14 \times 10^{-3}$ m between the plowed and unplowed field. Thus, single-point measurements of anemometric data may not account for $z_0$ over a range of spatial and temporal scales. Geometrically calculated $z_0$ using the Lettau method has shown to be an effective and more robust form of $z_0$ estimation compared to the anemometric method and also producing similar, estimated values. The anemometric, single-point measurements will also not account for the snow-covered area, which changes based on its inverse relation with $z_0$. However, SCA can be observed and estimated from satellite imagery or airborne lidar systems to create a more accurate estimation of $z_0$.

**Author Contributions:** The experiments were designed by D.J.K., S.R.F., and W.L.B.; D.J.K. collected the field data; D.J.K. and S.R.F. did the analysis; W.L.B. helped with instrumentation setup; I.O. prepared the code to compute the geometric roughness length; J.E.S. and S.R.F. wrote the paper with input from G.A.S.; S.R.F. created the figures.

**Funding:** This research has been partially supported by the National Science Foundation under Grant No. DMS-1615909.

**Acknowledgments:** We thank the Colorado State University Water Center and the Colorado Water Institute for their financial support, especially for the installation of the anemometric tower. The Warner College of Natural Resources provided the Faro Terrestrial Lidar Scanner. Thanks are due to Dr. Edgar Andreas who provided some very insightful discussions about the movement of air over snow. We thank the referees for their useful comments that helped to improve the presentation of the paper. Partial funding for SRF was provided by the National Science Foundation project *"Pattern Formation and Spatiotemporal Complex Dynamics in Extended Anisotropic Systems"* (PI Iuliana Oprea; award number DMS-1615909). Any use of trade, firm, or product names is for descriptive purposes only and does not imply endorsement by the U.S. Government. All data are available at Colorado State University by request to the corresponding author.

**Conflicts of Interest:** The authors declare no conflict of interest.

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
