# Peer review of "Geometric Versus Anemometric Surface Roughness for a Shallow Accumulating Snowpack"

_geosciences, doi:10.3390/geosciences8120463_

Round 1
Reviewer 1 Report
This clearly written paper addresses an interesting and novel set of research questions. It addresses an area of great uncertainty in energy balance modelling and presents novel findings that are substantiated by the data. I enjoyed reading the manuscript and have only minor recommendations for improvements and suggestions. My recommendation is that the paper can be considered for publication with only a minor revision.
Errors in wind tower z0 values
The “anemometric” z0 values (a new word for me!) are considered here as reference values – the case for this is made via citing knowledgeable sources (L89). However, I feel that there should be at least some acknowledgement of potential uncertainties in these wind tower derived values. One particular point that stands out for me is the effect of displacement height. How thick is this accumulated snow? What is the topographical variability? Where is the wind tower reference plane set? What thresholds were used to filter the data? What kind of anemometers/temp sensors were used, and what is their accuracy/error? I can see the benefit of using neutral stability conditions, but it is unclear to me how using wind speeds >4 m s-1 ensures this (L129).
Research questions
I can see the novelty of all three research questions. RQ (1) and (3) do overlap somewhat, and neither references the effect of plowing which is the focus of the first results figure. I don’t find the plowing aspect particularly well integrated into the rationale as it stands and wonder if a bit of tweaking is in order to improve clarity here.
Choice of patch size
As L93 mentions, the footprint area of the wind tower measurements depends on a number of factors. How was the 1000 m2 patch size decided upon? Could the patch have been limited to the area upwind of the tower? If that is a 50 m fetch (L122), then was it 20 m wide? What was the fetch? Was it homogeneous? Some basic justification would add clarity here. L126 says the field was plowed perpendicular to the prevailing wind direction. Were data filtered for wind direction? If not, did surface anisotropy have any effect on the anemometric/geometric values? What was the rough amplitude of troughs/furrows?
Survey data
A few more details would be nice when it comes to the topographic surveys. How many scan stations were used to cover any shadows? How were they stitched together? Presumably there were rather a few if ~30 x 30 m was surveyed on a flat field? Are the survey timings close to the 153 instances when anemometric data met stability criteria? Is this ‘solid surface’ (L137) a raster? What is the resolution of that? Were there any data gaps? Maybe a figure of the DEM would be helpful, if there is space.
Parameters of Lettau equation (and others)
Exactly how were the parameters of the Lettau equation computed? For instance, zh is the mean height of roughness elements (presumably above a set plane). Is this just the mean elevation of all cells when the whole field has been linearly detrended? Is there a directional component (important for frontal area)?
Snow accumulation vs Snowmelt
I find this to be an interesting finding emerging from the data. Accumulating snow has a different roughness value (for a given fractional cover) than melting snow. Have you identified a sort-of hysteretic effect? Also, it might be worth noting the smoothing effect of accumulating snow observed by Quincey et al. (2017) on a debris covered glacier, for comparison.
Quincey, D., Smith, M., Rounce, D., Ross, A., King, O. and Watson, C., 2017. Evaluating morphological estimates of the aerodynamic roughness of debris covered glacier ice. Earth Surface Processes and Landforms, 42(15), pp.2541-2553.
Figure 2 – the Lettau outlier is interesting – any comment to make on that?
Tiny Points
L111 – maybe this deserves to be a full blown equation as it doesn’t reproduce well here
L138 – CloudCompare should be cited I think
L195 – uncomfortable with the use of ‘actual value of z0’ as there will be errors in all measurements (see first main point above)
L242 – I feel like there should be some mention of the effect of data resolution on z0 (finer resolution = higher z0 – see Quincey et al./Miles et al.)
Figure 1 – if the x-axis is logarithmic scale, does this mean the bar widths cover different intervals? Is this appropriate?
Reviewer 2 Report
General comments
The manuscript presents a comparison between wind profile-derived and
geometry-based z0 estimates for a shallow snow surface and
examines the variability of z0 with snow covered area. The
study is original and covers a quite unexplored research area about
relations between aerodynamic and geometric roughness. This is of
valuable interest for models that would certainly benefit from such
observations to develop and evaluate parametrizations in this field.
The paper is concise and well written. My main suggestion is that the
methodology could be more thorough regarding the selection procedure
of the wind profiles and the determination of associated z0values. I recommend the paper as suitable for publication (with minor
revisions) providing the authors can address the following comments.
Specific comments
3. P2, L48: More generally changes in aerodynamic roughness have implications for turbulent transfer at the surface of the snowpack. An emphasis is made in the paper on the heat transfer, but nothing is said on the influence of z0 the representation of the wind field or on momentum transfer, while both play a crucial role in (re-)shaping the physical (geometric) surface roughness through erosion/building of roughness features. The latter is one of the main reasons for the temporal and spatial variability of surface roughness over windy areas permanently covered with snow (such as a large part of the Antarctic ice sheet (see Amory et al. 2017) or the accumulation zone of the Greenland ice sheet (see Smeets and van den Broeke 2008) and mountainous glaciers). I think the introduction should focus on these aspects as well.
P2, L66: Which methods? Computing z0from what? The above-mentioned methods deal with the geometric roughness, not z0.And the link with z0is far from being obvious. Here is an illustration of the importance of the wind direction when referring to z0. A given geometric roughness can correspond to a wide range of z0values according to the wind direction, and here is one of the main raisons complicating the definition of a direct relation between both variables. This aspect is not highlighted enough in the text.
More generally this paragraph (L52-66) can be a bit confusing, mainly because the difference between z0and the geometric roughness is not detailed enough. The sentence L66 actually follows on from the first sentence of the paragraph and its current position does not help to understand the key aspects of the problematic. Formally the aerodynamic roughness length z0is a mathematical integration constant, or more practically a turbulent quantity, while the surface roughness is a morphological characteristic of the snow surface. The aerodynamic roughness length z0over rough snow surfaces dependson the geometry of surface roughness features, but this geometry (or more precisely, the frontal area of roughness features) varies (temporally) depending on the wind direction and on the ability of the roughness features to reorient themselves according to the prevailing wind direction. This overall directional dependence accounts for the mention “aerodynamic” in the denomination of z0.These aerodynamic aspects should be discussed in the text (see Andreas and Claffey 1995; Amory et al. 2016).
My
recommendation would be to define z0and the
geometric roughness explicitly and soon enough in the introduction.
Eventually moving the first sentence of
the paragraph (L52) just before the sentence L66 will help to give
structure to the whole paragraph and to better expose the
problematic.
P3, L121-122: Monin-Obukhov similarity theory can only be applied to 'ideal' measurements, i.e., measurements performed at locations with long unobstructed upstream fetches during stationary conditions. The spatial homogeneity, or 'fetch' requirement, guarantees that unwanted effects caused (e.g., development of an internal boundary layer) by inhomogeneous upwind topographical conditions are negligible. In other words, surface-layer profiles are influenced by only one surface type. Can this requirement be expected to be met at your measurement site? It is not clear whether or not the meteorological tower is located sufficiently far from upstream obstacles. Is the west direction consistent with the mean wind direction in the area?
P3, L127: The mean wind estimate should be based on the longest practical interval that can be regarded as stationary. In turbulent boundary layers, averaging periods shorter than a few tens of minutes do not sufficiently smooth the usually occurring natural turbulent fluctuations of wind. Half-hourly or hourly (even one-hour-and-a-half) averages are usually employed in the literature. In practice, 10-min averages at minimum can satisfy this requirement. In any case, 5-min averages seem not long enough to discard the effect of transient eddies. Would using 10- ou 30- min averaged wind profiles affect the anemometric z0values?
P3, L128-129: You can easily compute a gradient Richardson number from the wind speed and temperature profiles rather than using a wind speed criterion only to ensure that statically neutral conditions are selected. This would strengthen the selection procedure.
About the requirement of stationarity, Joffre (1982) showed that periods with large temperature variations particularly have to be excluded when (quasi-) stationary conditions are to be considered (see Smeets and van den Broeke 2008 or Vignon et al. 2016 for selection criteria).
P4, L145: Is the very low amount of profiles (28 and 125) used to compute the statistics for each (plowed and unplowed) situation only due to the weak occurrence of wind speeds above 4 m/s?
P5, L29-130: Profile-based estimates of z0 can exhibit a large variation for an apparently homogeneous set of meteorological conditions, because natural conditions do not necessarily verify Monin-Obhukov requirements, particularly non-stationarity. How many profiles have been used to produce the anemometric z0 values used for comparison with each geometry-based z0 estimate? What is the variability that lies behind the average anemometric z0, or what (if you used only one value for comparison) if you consider the values before and after the selected profile? Could you provide an error estimate for anemometric z0 value? (Wilkinson (1984) provides a very straightforward and easy-computing way to do it)
P5, L168-175: Quantifying the variation of z0 according to the amount of snow-covered area or the unplowed or plowed situation would make the analysis more illustrative.
The variation of z0 with SCA seems very low in front of the variation of z0 with the plow-unplowed character of the field. This is an interesting result that deserves more attention in the text. How does the variability of z0 with SCA compare with the variability of anemometric z0 values around the date of acquisition? Is the wind direction identical for each comparison case?
P6, L186-188: By removing the Lettau’s constant of 0.5 the efficiency is significantly improved. Since there is no particular reason (and physical meaning) that justifies the use of another value for the constant rather than just removing it, can you obtain even better statistics by use of a particular value? In other words, what is the sensitivity of the efficiency to the constant?
P6,
L189-191: As you can only rely on anemometric z0 values
that are collected at a single point to evaluate the geometric z0,
I don’t see how the geometric data can account for spatial
variability in the roughness of the snow surface. More generally,
this conclusion remains valid in the context of your experiment but
still need to be verified in a different snowy environment. The
sensitivity to the value/removing of the Lettau’s constant can be
an illustration of the relative validity of the equation.
P8,L255-256: “more robust” compared to what? This conclusion could be nuanced if the differences between geometric and anemometric z0 values is of the order of the variability in the set of anemometric z0values used for comparison with each geometric z0.
Minor comments and typo suggestions
P2, L47: Give orders of magnitude.
P2,
L52: A coma is misplaced at the end of the sentence.
P2, L66: Use parenthesis for the date of publication.
P2, L53: Specify Millimeter-scale variations in snow surface “roughness” features.
P2, L67: Even 2-3 orders of magnitude (e.g., Andreas and Claffey 1995; Smeets and van den Broeke 2008; Amory et al. 2016; Vignon et al. 2016).
P2,
L84: See also Andreas (1995) and Amory et al. (2017)
P2, L87-88: This is also due to the directional dependance of z0and the lack of information on the spatio-temporal variability in the geometry of roughness features.
P3, L103: Specify that this is only valid through the hypothesis of stationarity and horizontal homogeneity.
P3, Eq. 1: Strictly, Phi is a function of the stability parameter z/L, not z0.
P3, L105-106: Prefer the formulation “z/L tends towards zero et phi can be neglected” rather than “z/L equals zero”.
P3, L110: It is hard to discriminate between indexes and true variables in the expression of the roughness area density. This equation could be inserted in the text just like the others.
Andreas E.L., 1995: Air-ice drag coefficients in the western weddell sea. 2. A model based on form drag and drifting snow. J Geophys Res 100(C3):4833–4843
Andreas, E.L., and Claffey K.J., 1995: Air-ice drag coefficients in the western weddell sea. 1. Values deduced from profile measurements. J Geophys Res 100(C3):4821–4831
Amory, C., Naaim-Bouvet, F,. Gallée, H., and Vignon, E., 2016: Brief communication: two well-marked cases of aerodynamic adjustment of sastrugi. The Cryosphere 10:1–8, 2016
Amory, C., Gallée, H., Naaim-Bouvet, F., Favier, V., Vignon, E., Picard, G., Trouvillez, A., Piard, L., Genthon, C., and Bellot, H., 2017: Seasonal Variations in Drag Coefficient over a Sastrugi-Covered Snowfield in Coastal East Antarctica, Boundary-Layer Meteorology, 164, 107–133
Joffre, S.M., 1982: Momentum and heat transfers in the surface layer over a frozen sea. Boundary-Layer Meteorol., 24:211–229
Smeets, C.J.P.P., and van den Broeke, M.R., 2008: Temporal and spatial variations of the aerodynamic roughness length in the ablation zone of the Greenland Ice Sheet. Boundary-Layer Meteorol 128(3):315–338
Vignon, E., Genthon, C., Barral, H., Amory, C., Picard, G., Gallée, H., Casasanta, G., and Argentini, S., 2016: Momentum- and Heat-FluxParametrization at Dome C, Antarctica: A Sensitivity Study, Boundary-Layer Meteorology, 162, 341–367
Wilkinson, R. H., 1984: A method for evaluating statistical errors associated with logarithmic velocity profiles. Geo-Mar Lett 3:49–52
